**Data Availability Statement:** Data are from British Columbia Perinatal database registry, which is available upon request from the British Columbia ministry of health. As the data contain sensitive

# The association between pre-pregnancy body mass index and perinatal death and the role of gestational age at delivery

Jeffrey N. Bone[1]*, K. S. Joseph[1,2], Chantal Mayer[1], Robert Platt[3], Sarka Lisonkova[1,2]

**1** Department of Obstetrics and Gynaecology, University of British Columbia and the Children's and Women's Hospital and Health Centre of British Columbia, Vancouver, BC, Canada, **2** School of Population and Public Health, University of British Columbia, Vancouver, BC, Canada, **3** Department of Epidemiology, Biostatistics, and Occupational Health, and of Pediatrics, McGill University, Montreal, Canada

* jeffrey.bone@cw.bc.ca

## Abstract

### Introduction

The pathophysiology behind the association between obesity and perinatal death is not fully understood but may be in part due to higher rates of pregnancy complications at earlier gestation amongst obese women. We aimed to quantify the proportion of perinatal deaths amongst obese and overweight women mediated by gestational age at stillbirth or live birth.

### Methods

The study included all singleton births at ≥20 weeks' gestation in British Columbia, 2004–2017, and excluded pregnancy terminations. The proportion of the association between BMI and perinatal death mediated by gestational age at delivery (in weeks) was estimated using natural effect models, with adjustment for potential confounders. Sensitivity analyses for unmeasured confounding and women missing BMI were conducted.

### Results

Of 392,820 included women, 20.6% were overweight and 12.8% obese. Women with higher BMI had a lower gestational age at delivery. Perinatal mortality was 0.5% (1834 pregnancies); and was elevated in overweight (adjusted odds ratio [AOR] = 1.22, 95% confidence interval [CI] 1.08–1.37) and obese women (AOR = 1.55, 95% CI 1.36–1.77). Mediation analysis showed that 63.1% of the association between obesity and perinatal death was mediated by gestational age at delivery (natural indirect effect AOR = 1.32, 95% CI 1.23–1.42, natural direct effect AOR = 1.18, 95% CI 1.05–1.32). Similar, but smaller effects were seen when comparing overweight women vs. women with a normal BMI. Estimated effects were not affected by adjustment for additional risk factors for perinatal death or sensitivity analyses for missing data.

patient information, and are owned by a third-party we are unable to share the analysis data publicly. Requests should be directed to PopData BC (dataaccess@popdata.bc.ca).

**Funding:** This research was supported by the Sick Kids Foundation (Grant # NI18-1272) and the Canadian Institute of Health Research. KSJ is supported by an Investigator award from the BC Children's Hospital Research Institute and SL is supported by a Scholar award from the Michael Smith Foundation for Health Research. RWP holds the Albert Boehringer I Chair in pharmacoepidemiology at McGill University. The funders had no role in study design, data collection and analysis, decision to publish, or preparation of the manuscript.

**Competing interests:** The authors have declared that no competing interests exist.

## Conclusion

Obese pregnancies have a higher risk of perinatal death in part mediated by a lower gestational age at delivery.

## Introduction

Obesity has increased dramatically in the past decades and become a growing public health concern [1]. These increases have also been observed amongst women of reproductive age [2]. In the USA, 26% of pregnant women were overweight and 25% were obese in 2014. High pre-pregnancy body mass index (BMI) is associated with many adverse maternal and perinatal outcomes [3–8]. For example, overweight women have a 36% higher risk of stillbirth than those of normal weight, and the risk is two-fold higher among obese women [3, 9]. Similar associations have been observed for neonatal and infant death [4, 6]. Obesity has thus become one of the most important preventable risk factors for adverse pregnancy outcomes.

While some obesity-related causes of fetal death are known, the exact pathophysiology behind the effects of obesity on perinatal death are not completely understood [10, 11]. High pre-pregnancy BMI is strongly associated with both spontaneous and clinician-initiated preterm birth, especially at extremely preterm gestation [8, 12, 13]. Among overweight and obese women, spontaneous preterm delivery or fetal compromise leading to clinician-initiated preterm delivery may be partly responsible for the increased risks of perinatal death. Better understanding of possible pathways between higher BMI and perinatal death, such as an increased frequency of pregnancy complications at early gestation (leading to stillbirth or preterm birth and neonatal death), may help elucidate pathological mechanisms underlying the increased risk and provide information for specific preventive recommendations.

The direct effect of obesity (i.e., the effect not arising through earlier gestation at delivery) is difficult to discern. As gestational age is on the causal pathway between obesity and perinatal death, the traditional analyses which include gestational age as a covariate in a multivariable regression model can yield spurious results [14]. Modern methods to assess mediation [15–18], however, can help quantify the fraction of the association between high BMI and perinatal mortality that is mediated through gestational age at birth. Mediation analysis with specific assumptions can partition the total effect of high BMI into the natural direct effect on perinatal death, and the mediated (or natural indirect) effect through gestational age at birth [19].

The primary aim of our study was to quantify the proportion of the association between pre-pregnancy BMI and perinatal death that is mediated through differences in gestational age at birth.

## Methods

### Data source and study population

The study included women with a singleton live birth or stillbirth in British Columbia between April 1, 2004 and March 31, 2017. The cohort was obtained from the British Columbia Perinatal Database Registry (BCPDR) [20] which contains individual-level data on >99% of all live births and stillbirths in British Columbia. The registry captures nearly 300 pre-pregnancy, obstetric and neonatal data elements abstracted by trained personnel from the hospital charts and from delivery records of midwives attending home births. The information included up to 25 diagnostic fields (coded using the International Classification of Diseases and Related

Health Problems, Canadian version, ICD-10-CA) and 20 procedure fields (coded using Canadian Classification of Interventions) related to the hospitalization or home birth for the mother and the infant. Additionally, the BCPDR includes data on hospital readmissions for mothers and infants within 42 and 28 days, respectively. Prior validation studies have found high sensitivity and specificity of collected data regarding major pregnancy events and risk factors compared with medical charts [21].

Births at <20 weeks' gestation and late pregnancy terminations were excluded. Birth records with missing information on BMI or relevant confounders were excluded from the primary analyses (S1 Fig).

## Exposure, outcome and confounders

Self-reported pre-pregnancy weight and height were used to calculate BMI, categorized as follows: underweight (BMI <18.5 m/kg$^2$), normal (18.5m/kg$^2$ ≤ BMI < 25m/kg$^2$), overweight (25m/kg$^2$ < BMI < 30m/kg$^2$) and obese (BMI ≥30 m/kg$^2$). The primary outcome was perinatal death, defined as either a stillbirth or neonatal death before hospital discharge. Gestational age at delivery, in weeks, was determined from a standard algorithm incorporating early ultrasound (before 20 weeks), last menstrual period, newborn clinical exam, and chart-documented gestational age [20]. Information on maternal and pregnancy characteristics included: maternal age, parity, chronic hypertension, assisted conception, self-reported use of drugs, alcohol or cigarettes during pregnancy, asthma, pre-pregnancy diabetes, chronic maternal conditions, year of delivery, sex of fetus, congenital anomalies and prior stillbirth, low birth weight infant, or caesarean delivery. Suspected intrauterine growth restriction (IUGR) was identified by prenatal ultrasound. Chronic conditions included cardiovascular diseases, chronic renal and hepatic conditions, and systemic lupus erythematosus (S1 Table).

## Statistical analysis

All demographic and obstetric history data were summarised by BMI category. Perinatal death and stillbirth rates were expressed per 10,000 total births, while neonatal death rates were expressed per 10,000 live births. Gestational age-specific perinatal death rates were expressed per 10,000 fetuses-at-risk [22].

## Mediation analysis

We conducted a mediation analysis under a potential outcomes framework [23] to determine the proportion of the association between BMI and perinatal death mediated by gestational age at stillbirth or live birth. This analysis aimed to quantify how much of the total association between BMI and the outcome was direct and how much was mediated through differences in the distribution of gestational age at birth. The natural direct effect represents the influence of pre-pregnancy BMI on perinatal death assuming no difference in gestational age at delivery between women with normal vs high pre-pregnancy BMI. Conversely, the natural indirect effect, i.e., the mediated effect, measures the impact that higher BMI has on earlier gestation at delivery, and thus perinatal death. Namely, the indirect effect contrasts the expected perinatal death rate assuming that timing of all deliveries in women with high BMI was independent (i.e. their gestational age at delivery was what it would be if they had normal BMI) versus the perinatal death rate observed among women with high BMI (given their actual gestational age at delivery, formal definitions in S1 Appendix). Standard errors were based on the robust 'sandwich' estimator [24].

Unadjusted and adjusted mediation analyses were performed, with the latter including chronic hypertension, smoking, substance/alcohol use, prior stillbirth, prior preterm birth,

parity, maternal age, year of birth, chronic diseases, and asthma as covariates. These confounders were thought to be associated with pre-pregnancy BMI and perinatal death (i.e., the exposure and the outcome), or with gestational age at delivery and perinatal death (i.e., the mediator and the outcome; S2 Fig) [25]. We did not have information about timing of chronic hypertension onset and because it may (for some women) be on the causal pathway between obesity and perinatal death, we performed the same analyses without adjustment for chronic hypertension. We found no indication of interaction between BMI and gestational age at delivery, and therefore mediation models did not include this term [26].

Causal interpretations of these mediation analyses require strong assumptions. Firstly, that there are no unmeasured confounders of the relationships between pre-pregnancy BMI and gestational age, pre-pregnancy BMI and perinatal death, and gestational age and perinatal death. Secondly, that none of the confounders of the mediator/outcome relationship are caused by the exposure [17, 23, 27]. In this case, this assumes that there are no additional mediators between increased BMI and perinatal death that also effect gestational age at delivery.

### Sensitivity analyses

To assess the robustness of our results, we conducted sensitivity analyses to determine the amount of unmeasured confounding (E-value) in the association between the gestational age at delivery and perinatal death required to explain away the estimated natural direct and indirect effects [28, 29]. To assess the impact of women missing BMI on our results we conducted multiple imputation by chained equations (10 datasets). In each imputed data set the natural direct and indirect effects and proportion mediated were estimated, then pooled using Rubin's rules [30]. Additionally, pre-pregnancy diabetes and congenital anomalies were considered as possible exposure induced mediator-outcome confounders in sensitivity analyses. In this last analyses, natural effects as defined above were not identifiable, but analogous 'interventional' effects were (see S1 Appendix for details) [31–33].

Finally, we repeated our primary analysis for each of stillbirth, antepartum stillbirth and neonatal death separately to determine if the relationship observed for the primary outcome differed. Due to small numbers, we were unable to conduct this analysis for intrapartum stillbirth.

All analyses were carried out using R version 3.5.3 [34], with the *medflex* [35] package used for mediation analyses.

Written ethics approval was obtained from the University of British Columbia—Children's and Women's Hospital and Health Centre of British Columbia Research Ethics Board (H18-03154). All data were fully anonymized prior to data access or analyses. Data were from administrative registry and individual participant consent was not required.

## Results

### Study population

A total of 547,401 singleton live births and stillbirths occurred in British Columbia between April 1, 2004, and March 31, 2017 (excluding late terminations). After exclusion of women with births prior to 20 weeks' gestation and those with unknown gestation, the study population included 546,675 (99.9%) women. Pre-pregnancy BMI was missing in 153,855 women (28.1%; S1 Fig), and the main analyses included 392,820 women. Of these, 50,352 (12.8%) were classified as obese, 81,065 (20.6%) overweight, 237,726 (60.6%) normal weight and 23,677 (6.0%) underweight. Obese women were more likely to be multiparous, smoke during pregnancy, have chronic and gestational hypertension or diabetes, and a prior caesarean delivery as compared with women with normal BMI (Table 1). Gestational age differences were also

**Table 1. Demographic characteristics, pre-pregnancy, and pregnancy morbidity of women by pre-pregnancy BMI\* category.**

| Demographics | Underweight N = 23,677 Number (%) | Normal BMI N = 237,726 Number (%) | Overweight N = 81,065 Number (%) | Obese N = 50,352 Number (%) |
|---|---|---|---|---|
| Maternal age (years) | | | | |
| <20 | 1013 (4.3) | 5595 (2.4) | 1549 (1.9) | 742 (1.5) |
| 20–34 | 18799 (79.4) | 175690 (73.9) | 59852 (73.8) | 38277 (76.0) |
| 35–39 | 3226 (13.6) | 46396 (19.5) | 15912 (19.6) | 9369 (18.6) |
| ≥40 | 639 (2.7) | 10045 (4.2) | 3752 (4.6) | 1964 (3.9) |
| Nulliparous | 13509 (57.1) | 121285 (51.0) | 36087 (44.5) | 20564 (40.8) |
| Year of delivery | | | | |
| 2004–2006 | 4944 (20.9) | 47161 (19.8) | 15110 (18.6) | 8682 (17.2) |
| 2007–2009 | 5188 (21.9) | 51187 (21.5) | 17455 (21.5) | 10485 (20.8) |
| 2010–2012 | 5364 (22.7) | 55878 (23.5) | 19325 (23.8) | 12066 (24.0) |
| 2013–2014 | 3827 (16.2) | 39065 (16.4) | 13353 (16.5) | 8654 (17.2) |
| 2015–2017 | 4354 (18.4) | 44435 (18.7) | 15822 (19.5) | 10465 (20.8) |
| **Pre-pregnancy conditions** | | | | |
| Prior stillbirth | 96 (0.4) | 1396 (0.6) | 675 (0.8) | 539 (1.1) |
| Prior low birth weight infant | 477 (2.0) | 3710 (1.6) | 1460 (1.8) | 975 (1.9) |
| Prior caesarean section | 2041 (8.6) | 28576 (12.0) | 13626 (16.8) | 10604 (21.1) |
| Chronic disease | 50 (0.2) | 556 (0.2) | 158 (0.2) | 102 (0.2) |
| Asthma | 126 (0.5) | 1529 (0.6) | 639 (0.8) | 478 (0.9) |
| IVF\*\* | 284/15139 (1.9) | 3802/156081 (2.4) | 1422/54676 (2.6) | 834/34967 (2.4) |
| Pre-pregnancy hypertension | 35 (0.1) | 748 (0.3) | 706 (0.9) | 1220 (2.4) |
| Pre-pregnancy diabetes mellitus | 480 (2.0) | 4915 (2.1) | 2100 (2.6) | 1748 (3.5) |
| **Pregnancy complications** | | | | |
| Gestational diabetes | 1782 (7.5) | 22117 (9.3) | 12696 (15.7) | 12375 (24.6) |
| Gestational hypertension | 1323 (5.6) | 15162 (6.4) | 8483 (10.5) | 8202 (16.3) |
| Placental previa | 158 (0.7) | 1666 (0.7) | 564 (0.7) | 276 (0.5) |
| Placental disorders (excl. previa) | 168 (0.7) | 1758 (0.7) | 566 (0.7) | 324 (0.6) |
| Preeclampsia | 64 (0.3) | 978 (0.4) | 616 (0.8) | 763 (1.5) |
| Smoking during pregnancy\* | 2224 (9.4) | 16044 (6.7) | 7102 (8.8) | 5965 (11.8) |
| Alcohol use during pregnancy\* | 196 (0.8) | 2093 (0.9) | 828 (1.0) | 635 (1.3) |
| Drug use during pregnancy\* | 957 (4.0) | 6503 (2.7) | 2556 (3.2) | 1925 (3.8) |
| Suspected IUGR\*\*\* | 998 (4.2) | 4785 (2.0) | 1166 (1.4) | 740 (1.5) |
| **Fetal/infant characteristics** | | | | |
| Gestational age at delivery (weeks) | | | | |
| 20–27 | 93 (0.4) | 757 (0.3) | 331 (0.4) | 279 (0.6) |
| 28–33 | 247 (1.0) | 1833 (0.8) | 758 (0.9) | 574 (1.1) |
| 34–36 | 1692 (7.1) | 14797 (6.2) | 5512 (6.8) | 4004 (8.0) |
| 37–38 | 7803 (33.0) | 72000 (30.3) | 25166 (31.0) | 17037 (33.8) |
| 39–41 | 13583 (57.4) | 145533 (61.2) | 48302 (59.6) | 27831 (55.3) |
| ≥42 | 259 (1.1) | 2806 (1.2) | 996 (1.2) | 627 (1.2) |
| Male fetus | 11970 (50.6) | 122220 (51.4) | 41489 (51.2) | 25849 (51.3) |
| Congenital anomaly (any)\*\*\*\* | 51 (0.2) | 618 (0.3) | 216 (0.3) | 150 (0.3) |

\*Self-reported

\*\*In-vitro-fertilization, information available only in pregnancies delivered in 2008/2009 onwards

\*\*\* Intrauterine growth restriction suspected on antenatal ultrasound (i.e., prior to delivery; antenatal ultrasound does not detect all growth-restricted fetuses that are born small-for-gestational age)

\*\*\*\* Only among live births, diagnosed at birth.

**Table 2. Perinatal mortality by pre-pregnancy BMI category.**

|  | Underweight | Normal BMI | Overweight | Obese |
|---|---|---|---|---|
|  | N = 23,677 | N = 237,726 | N = 81,065 | N = 50,352 |
| Stillbirth (n, per 10,000 total births) | 64 (27.0) | 632 (26.5) | 270 (33.3) | 225 (44.7) |
| Stillbirth timing |  |  |  |  |
| Intrapartum (n,%) | 6 (9.4%) | 48 (7.6%) | 26 (9.6%) | 19 (8.4%) |
| Antepartum (n,%) | 51 (79.7%) | 520 (82.3%) | 224 (83.0%) | 182 (80.9%) |
| Unknown (n,%) | 7 (10.9%) | 64 (10.1%) | 20 (7.4%) | 24 (10.7%) |
| Neonatal death (n, per 10,000 live births) | 49 (20.8) | 346 (14.6%) | 142 (17.6) | 106 (21.2) |
| Perinatal death (n, per 10,000 total births) | 113 (47.7) | 978 (41.1) | 412 (50.8) | 331 (65.7) |
| Stillbirth as a proportion of perinatal death | 56.6% | 64.6% | 65.6% | 68.0% |

evident: the preterm birth rate was 7.3% among women with a normal BMI, and 8.1% and 9.7%, among overweight and obese women, respectively.

## Main outcomes

Women who were underweight, overweight and obese had higher rates of stillbirth, neonatal death and perinatal death (Table 2). In all BMI categories, the majority of perinatal deaths were antepartum stillbirths. The rates of antepartum stillbirth were similar between women who were underweight prior to pregnancy and those with normal BMI. Gestational age-specific perinatal death rates (per 10,000 fetuses-at-risk) increased with gestational age in women in all BMI categories and were generally higher in women with non-normal BMI (Fig 1).

There was no strong evidence of an association between low BMI (underweight) and perinatal mortality (OR = 1.16, 95% CI: 0.96, 1.40). However, there was a positive unadjusted association between overweight and obese and perinatal death (OR = 1.24, 95% CI 1.10, 1.39 and OR = 1.61, 95% CI: 1.41, 1.83, respectively) and this association was only slightly attenuated after adjustment for confounders (Table 3).

## Mediation analyses

Mediation analyses showed that the indirect effects of BMI on perinatal death through gestational age at delivery were larger than the direct effects (Table 3). Half of the association between overweight BMI and perinatal death was mediated through gestational age at delivery (50.4%). The proportion of the association between obesity and perinatal death mediated by gestational age at delivery was 63.1% (natural direct effect AOR = 1.18, 95% CI: 1.05, 1.32; vs. natural indirect effect = 1.32, 95% CI: 1.23, 1.42). Estimates of the total, direct and indirect effects were similar between unadjusted and adjusted results. Further analyses adjusting for pre-pregnancy diabetes and congenital anomalies and those excluding chronic hypertension from adjustment set of covariates did not impact the estimates.

## Sensitivity analyses

Sensitivity analyses indicated that to explain away the statistical significance of the indirect effect of obesity on perinatal death by unmeasured confounding, such a confounder(s) (associated with gestational age at delivery and perinatal death) would have to increase the rate of perinatal death and early gestational age at delivery by at least 75%. Similarly, such an unmeasured confounder would have to increase theses rates by 28% to explain away the direct effect. For overweight women unmeasured confounding would have to increase the rate of perinatal

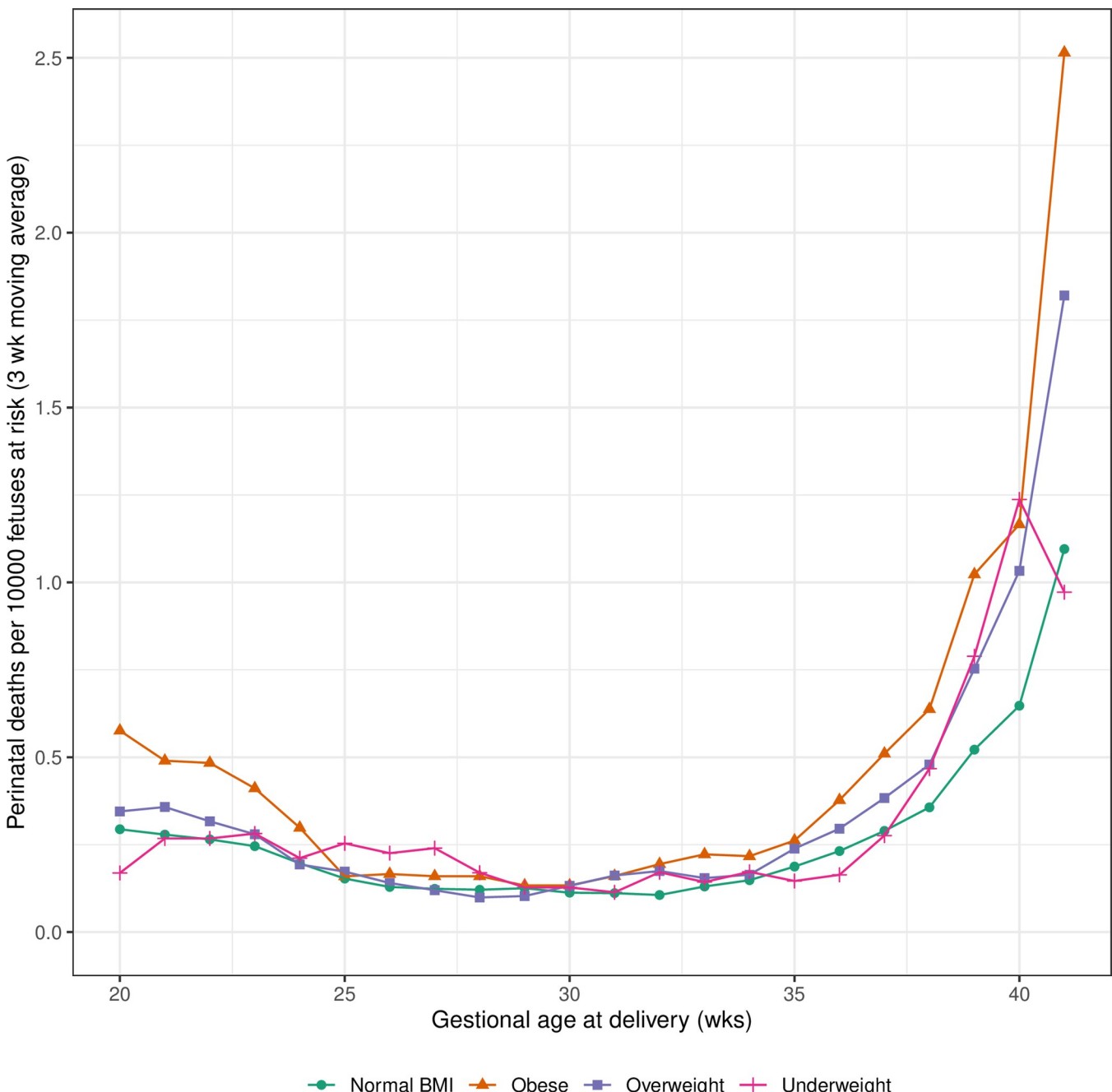

**Fig 1. Gestational age specific perinatal death rates per 10,000 fetuses at risk by BMI class.** Data are smoothed with three week moving averages.

death and early gestational age at delivery by 25% to explain away the indirect effect (S2 Table).

There were no substantial differences in demographic characteristics between women with complete vs missing information about pre-pregnancy BMI, with the exception of year of delivery (higher proportion of missing BMI in earlier years), nulliparity, and previous caesarean delivery (S3 Table). Those with missing BMI had higher rates of perinatal death (99.5 per 10,000 vs 46.6 per 10,000 total births among those without missing information). Further, a

**Table 3. Mediation analyses of BMI on perinatal death through gestational age at delivery.**

| BMI Category | Underweight | Normal BMI | Overweight | Obese |
|---|---|---|---|---|
| **Unadjusted** | | | | |
| Total effect (OR) | 1.16 (0.96, 1.40) | Ref | 1.24 (1.10, 1.39) | 1.61 (1.41, 1.83) |
| Natural direct | 1.03 (0.87, 1.21) | Ref | 1.09 (0.99, 1.20) | 1.14 (1.02, 1.28) |
| Natural indirect | 1.13 (1.03, 1.24) | Ref | 1.13 (1.07, 1.20) | 1.41 (1.31, 1.51) |
| Proportion mediated (indirect effect) | - | - | 59.7% | 72.0% |
| **Adjusted*** | | | | |
| Total effect (AOR-2) | 1.17 (0.96, 1.41) | Ref | 1.22 (1.08, 1.37) | 1.55 (1.36, 1.77) |
| Natural direct | 1.01 (0.86, 1.19) | Ref | 1.10 (1.00, 1.22) | 1.18 (1.05, 1.32) |
| Natural indirect | 1.15 (1.05, 1.26) | Ref | 1.11 (1.04, 1.17) | 1.32 (1.23, 1.42) |
| Proportion mediated (indirect effect) | - | - | 50.4% | 63.1% |
| **Adjusted**** | | | | |
| Total effect (AOR-1) | 1.17 (0.96,1.41) | Ref | 1.22 (1.09,1.37) | 1.57 (1.38,1.80) |
| Natural direct*** | 1.01 (0.86,1.20) | Ref | 1.10 (1.00,1.21) | 1.16 (1.03,1.30) |
| Natural indirect*** | 1.15 (1.04,1.26) | Ref | 1.11 (1.05,1.18) | 1.36 (1.26,1.46) |
| Proportion mediated (indirect effect) | - | - | 53.2% | 67.4% |
| **Adjusted***** | | | | |
| Total effect (AOR-3) | 1.17 (0.96,1.41) | Ref | 1.22 (1.09,1.37) | 1.55 (1.36,1.77) |
| Natural direct**** | 1.01 (0.86,1.19) | Ref | 1.11 (1.00,1.22) | 1.18 (1.05,1.33) |
| Natural indirect**** | 1.15 (1.05,1.27) | Ref | 1.10 (1.04,1.17) | 1.31 (1.22,1.41) |
| Proportion mediated (indirect effect) | - | - | 49.0% | 61.6% |

\* adjusted for chronic hypertension, smoking, substance/alcohol use, prior stillbirth, prior preterm birth, parity, maternal age, year of birth, chronic diseases, asthma

\*\* adjusted for all above except chronic hypertension.

\*\*\* adjusted for all above and pre-pregnancy diabetes and congenital anomalies

\*\*\*\* interpreted as 'interventional effects, see S1 Appendix.

higher proportion of the perinatal deaths among women with missing BMI were stillbirths (71% vs 64%). The results using multiple imputation for missing BMI were similar to the complete-case analyses, albeit with a lower proportion of the total effect mediated (S4 Table).

Compared to the primary analyses for perinatal death, analyses restricted to stillbirth led to slightly higher odds ratios overall, while analyses for neonatal death were attenuated. Direct effects were higher, and mediation were lower for stillbirth than for neonatal death, where the total effect was entirely indirect (S5 Table). Despite this, confidence intervals for all estimates were consistent with the primary analyses (S5 Table).

## Discussion

Our population-based study shows that high pre-pregnancy BMI is positively associated with perinatal death: overweight women had a 22% (95% CI 8–37%) higher adjusted perinatal death rate compared with women with a normal BMI, while obese women had a 55% (95% CI 36–77%) higher adjusted perinatal death rate. Overweight and obese women delivered at a lower mean gestational age, and mediation analyses showed that 63% of the association between pre-pregnancy obesity and perinatal death was mediated by such gestational age differences.

Elevated rates of perinatal death among overweight and obese women have been described previously. Our findings of 55% increased risk of perinatal death in obese women is similar to previous reports [8, 36]. However, some studies have found more than 2-fold elevated risk

associated with pre-pregnancy obesity [4, 37, 38], which could reflect differences in study populations. Also noteworthy, differences in the proportion of obese women in our population (12.8%) was lower than in other high-income countries [4, 36], and in the Canadian population as a whole [39]. Our study showed that the stillbirth rate and the proportion of stillbirths among perinatal deaths increased with increasing pre-pregnancy BMI. This could reflect clinical challenges in antenatal fetal monitoring in women with high pre-pregnancy BMI [40].

The mediating effect of gestational age at delivery likely applies to both components of perinatal death, viz., stillbirth and neonatal death. Overweight and obese women have increased rates of medically-indicated delivery at preterm gestation, and increased rates of spontaneous preterm delivery, especially at extremely preterm gestation (<28 weeks) compared with women with normal BMI [13]. Since preterm infants have immature organ systems, such births experience higher rates of neonatal death; the association between pre-pregnancy BMI and neonatal death is therefore likely to be, in large part, mediated through such early delivery. The mediating effect of gestational age on stillbirth is less clear though it is known that some causes of fetal death, such as infection, are significantly more frequent at early gestation [41]. It is noteworthy that the indirect effect of pre-pregnancy obesity is substantial and its magnitude (63.1%) is larger than the indirect effect of pre-pregnancy overweight (50.4%); stillbirths constitute a large proportion of perinatal deaths, and the stillbirth proportion is slightly larger among obese women compared with overweight women. This suggests that pregnancy complications at early gestation, which increase the risk of antepartum stillbirth, play an important role in mediating the association between high pre-pregnancy BMI and perinatal death.

Most previous studies have only reported the total effects of high BMI on perinatal death, which obscures the mechanisms underlying the effects of pre-pregnancy BMI. One study [38] that estimated the direct effect of obesity by adjusting for gestational age in multivariable analysis, showed a stronger adjusted association between obesity and perinatal death and implied negative confounding by preterm delivery. Such paradoxical results can occur when variables in the causal pathway between exposure and outcome are included in causal models [14] in the presence of confounders between these intermediate variables and the outcome. To avoid this form of collider-stratification bias, we used mediation analysis to partition natural direct and indirect effects.

Our estimated mediated effect provides insight into the possible mechanism by which adverse perinatal outcomes are disproportionately observed in overweight and obese women. Maternal complications of pregnancy, e.g., gestational hypertension and preeclampsia, are more frequent in overweight and obese women [41]. Higher rates of pregnancy complications at preterm gestation result in higher rates of preterm live birth and stillbirth thereby contributing to the indirect effect of obesity. Conversely, a fraction of stillbirths that occur among normal, overweight, and obese women have similar gestational age distributions, although they occur at higher rates among overweight and obese women. These stillbirths are responsible for the direct effect of obesity on perinatal death. Causes of death that underlie such direct effects likely include placental disease (e.g. uteroplacental insufficiency, placenta previa), fetal structural and genetic anomalies, and umbilical cord abnormalities [10]. Other conditions such as severe preeclampsia, contribute to both direct and indirect effects of obesity by leading to higher rates of stillbirth at any gestational age and to higher rates of clinician-initiated preterm delivery (sometimes resulting in neonatal death). Some maternal and fetal complications of pregnancy can be represented through complex causal pathways between high pre-pregnancy BMI and perinatal death and pose a challenge to evaluating the assumptions required for mediation analysis.

Previous studies have indicated that the increased rate of preterm birth in obese women is in large part due to an increase in iatrogenic deliveries [12, 13]. The indirect effect of BMI on

perinatal death is thus related to clinical management and optimal of timing of delivery in these women [42]. While a physician-indicated preterm delivery may prevent fetal death, it may increase the risk of neonatal death. This issue poses major challenges not only for health care practitioners but also for research methodologists [43]. Physician-indicated delivery is a competing risk (or a competing outcome) for stillbirth. For this reason, we used a composite of stillbirth and neonatal (perinatal death). However, further studies may be needed to disentangle the pathways through which these competing risks are realized.

The causal interpretation of these results requires an assumption of no unmeasured confounding throughout, and the absence of any confounder of gestational age at delivery and perinatal death that are effected by BMI [26]. The comprehensiveness of our data set makes it unlikely that clinical confounders important to our studied relationships were excluded. However, lack of socioeconomic data may represent a source of unmeasured confounding. To assess this, we performed sensitivity analyses to assess the impact of such unmeasured confounders and found that our results were reasonably robust. On the other hand, it is likely that there are confounders between gestational age at delivery and perinatal death are affected by pre-pregnancy BMI. In this case, if these exposure induced mediator-outcome confounders are adjusted for, one can still identify 'interventional' direct and indirect effects, which correspond to changing the distribution of the mediator in the exposed/unexposed rather than changing individual's mediator values (see S1 Appendix for a detailed discussion) [31–33]. We included two such possible confounders in a sensitivity analysis (pre-pregnancy diabetes and congenital anomalies) and found little difference in estimates. Despite this, additional BMI induced confounders of gestational age and perinatal death may exist and could impact our findings. The validity of these types of assumptions is uncertain in most observational studies, and particularly those with complex causal pathways such as ours.

Furthermore, there has been considerable debate about the relevance of 'effects' of exposures such as pre-pregnancy BMI as various conditions, such as sedentary lifestyle, high calorie food intake and medical or genetic conditions can lead to high BMI and all may have different effects on outcomes (in such situations, more general effects corresponding to reducing BMI by a 'random intervention' can be identified, see S1 Appendix for details) [44–48]. Despite this, BMI remains an important clinical indicator of risk in pregnancy, and exploration of the reasons for this increased risk remain of value, even without directly linked interventions. In this study, we examined the relationship of obesity and perinatal death with respect to the 'average' population prevalence of various causes of high BMI [46]. This limits the interpretation of our findings to a more explanatory nature in that the results cannot be linked to the possible effect of a specific obesity related intervention.

Strengths of our study include the large population-based nature of our data and comprehensive information on possible confounders. Our study also has some limitations. First, the direct and indirect effects estimated require strong assumptions to be considered causal as discussed above. In particular, it is possible that there are common risk factors of both early delivery and perinatal death that are also affected by obesity. It would be of interest in future work to include such variables in models accounting for multiple mediators to assess their joint mediated effect [49]. Second, as a mediator, we used gestational age at stillbirth (i.e., gestational age at delivery of a dead fetus) which occurs after the actual in-utero fetal death. This is important because from the temporal point of view, gestational age at stillbirth cannot be a mediating factor between pre-pregnancy BMI and in-utero fetal death. Instead, gestational age at stillbirth approximates the time of fetal death and the intra-uterine environment leading to fetal demise. While gestational age at fetal death may have been preferable in this context, the difference between gestational age at fetal death and at stillbirth is usually a matter of hour and days (rarely a week or more), especially among singleton pregnancies. Hence, this limitation is

unlikely to have significantly impacted our results. Thirdly, the information about pre-pregnancy BMI (weight and height) was self-reported, possibly leading to some misclassification, and missing in 28% of women, possibly leading to a selection bias. As for the latter, sensitivity analyses using multiple imputation for missing BMI yielded results similar with primary analyses. Further, we did not adjust for correlation between subsequent pregnancies of the same woman during the study period, as the information was not available.

## Conclusion

Pre-pregnancy obesity carries increased risk of perinatal mortality. Our mediation analysis results suggest that earlier gestational age at delivery may be implicated in many of these obesity-related perinatal deaths. Timely obstetric intervention coupled with access to neonatal intensive care earlier in gestation may further mitigate the risk of neonatal death among infants in these women. To better inform the pregnancy management in obese women, further studies should continue to disentangle the causal pathways under which obesity increases the risk of perinatal death, including, for example gestational diabetes and other obesity-related pregnancy complications.

## Supporting information

**S1 Checklist. STROBE 2007 (v4) statement—checklist of items that should be included in reports of** *cohort studies***.**
(DOCX)

**S1 Table. ICD-10 codes definitions of relevant variables.**
(DOCX)

**S2 Table. The amount of unmeasured confounding (as a risk ratio) between mediator and outcome needed to explain away total, direct and indirect effects.** Estimates for underweight estimates are not presented as the total effects were not significant. The e-value is also not estimated when the effect is not significant.
(DOCX)

**S3 Table. Comparison of women with and without complete BMI data.**
(DOCX)

**S4 Table. Mediation analyses of BMI on perinatal death through gestational age at delivery after multiple imputation (n = 10) of missing BMI values.**
(DOCX)

**S5 Table. Mediation analyses for stillbirth and neonatal death.**
(DOCX)

**S1 Fig. Study population.**
(DOCX)

**S2 Fig. Directed acyclic graph representing assumed causal structure for causal mediation analysis.** (C and L represent two sets of confounders: C confounders between the exposure and the outcomes; L confounders between the mediator and the outcome).
(DOCX)

**S1 Appendix.**
(PDF)

## Acknowledgments

All inferences, opinions and conclusions drawn in this paper are those of the authors and do not reflect the opinions or policies of the Data Stewards.

## Author Contributions

**Conceptualization:** Jeffrey N. Bone, K. S. Joseph, Sarka Lisonkova.

**Formal analysis:** Jeffrey N. Bone.

**Investigation:** Sarka Lisonkova.

**Methodology:** Jeffrey N. Bone, K. S. Joseph, Robert Platt, Sarka Lisonkova.

**Supervision:** Sarka Lisonkova.

**Writing – original draft:** Jeffrey N. Bone.

**Writing – review & editing:** K. S. Joseph, Chantal Mayer, Robert Platt, Sarka Lisonkova.

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
