## [Decision Letter · Decision Letter 0]

4 Jan 2022

PONE-D-21-29394The association between pre-pregnancy body mass index and perinatal death and the role of gestational age at deliveryPLOS ONE

Dear Dr. Bone,

Thank you for submitting your manuscript to PLOS ONE. After careful consideration, we feel that it has merit but does not fully meet PLOS ONE’s publication criteria as it currently stands. Therefore, we invite you to submit a revised version of the manuscript that addresses the points raised during the review process.

We look forward to receiving your revised manuscript.

Kind regards,

Angela Lupattelli, PhD

Academic Editor

PLOS ONE

https://journals.plos.org/plosone/s/file?id=ba62/PLOSOne_formatting_sample_title_authors_affiliations.pdf”

2. Please provide additional details regarding participant consent. In the ethics statement in the Methods and online submission information, please ensure that you have specified what type you obtained (for instance, written or verbal, and if verbal, how it was documented and witnessed). If your study included minors, state whether you obtained consent from parents or guardians. If the need for consent was waived by the ethics committee, please include this information."

If you are reporting a retrospective study of medical records, archived samples or third party data, please ensure that you have discussed whether all data were fully anonymized before you accessed them and/or whether the IRB or ethics committee waived the requirement for informed consent. If patients provided informed written consent to have data from their medical records used in research, please include this information.

“This research was supported by the Sick Kids Foundation (Grant # NI18-1272) and the Canadian Institute of Health Research. KSJ is supported by an Investigator award from the BC Children’s Hospital Research Institute and SL is supported by a Scholar award from the Michael Smith Foundation for Health Research. RWP holds the Albert Boehringer I Chair in pharmacoepidemiology at McGill University.

All inferences, opinions and conclusions drawn in this paper are those of the authors and do not reflect the opinions or policies of the Data Stewards.”

6. We note that you have indicated that data from this study are available upon request. PLOS only allows data to be available upon request if there are legal or ethical restrictions on sharing data publicly. For more information on unacceptable data access restrictions, please see http://journals.plos.org/plosone/s/data-availability#loc-unacceptable-data-access-restrictions.

7. Please include a caption for figure 1.

Reviewers' comments:

Reviewer's Responses to Questions

**Comments to the Author**

1. Is the manuscript technically sound, and do the data support the conclusions?

Reviewer #1: Yes

Reviewer #2: Yes

2. Has the statistical analysis been performed appropriately and rigorously? 

Reviewer #1: Yes

Reviewer #2: Yes

3. Have the authors made all data underlying the findings in their manuscript fully available?

Reviewer #1: No

Reviewer #2: Yes

4. Is the manuscript presented in an intelligible fashion and written in standard English?

Reviewer #1: Yes

Reviewer #2: Yes

5. Review Comments to the Author

Reviewer #1: This is an interesting analysis of maternal pre-pregnancy BMI and perinatal death from a large cohort study in British Columbia including 392820 women. The study found increased risk of perinatal death among overweight and obese women, with RRs of 1.22 and 1.55, respectively. A large part of this association (63.1%) was mediated by gestational age at delivery. These findings are interesting and I only have a few minor comments below.

Please add number of perinatal deaths to the abstract.

What is the rationale for adjusting for hypertension? Hypertension could also be on the causal pathway between maternal BMI and perinatal death. I would suggest to exclude hypertension from the multivariable model or alternatively present an additional model without hypertension, but with the other covariates.

Line 243-244: Has this statement been misplaced?

Line 295: stillbirths? Plural s

Did you try testing hypertension as a potential mediator in this dataset?

Line 373: …..the risk of neonatal death among infants of these women.

Please, would you be able to add analyses of stillbirth (overall, and intrapartum and antepartum), and neonatal death as well, since you have the data in Table 2? At least in an online supplement. That could have been helpful for the overall evidence base.

Considering the large study population, have you considered doing sensitivity analyses using finer BMI categories or at least looking at severe obesity separately from grade 1 obesity? Perhaps this could be added in an online supplement.

Reviewer #2: The authors wanted to examine the possibility that earlier delivery of obese pregnancies may be attributable to increased stillbirth and neonatal death. But I think the cause and the mechanism of stillbirth and neonatal death are different. I don't suggest discussing the two together. And pregnancy complications such as GDM,Pregnancy induced hypertension，ICP can lead to stillbirth.So they should be excluded.

6. PLOS authors have the option to publish the peer review history of their article (what does this mean?). If published, this will include your full peer review and any attached files.

Reviewer #1: No

Reviewer #2: No

---

## [Author Response · Author response to Decision Letter 0]

3 Feb 2022

Reviewer #1: This is an interesting analysis of maternal pre-pregnancy BMI and perinatal death from a large cohort study in British Columbia including 392820 women. The study found increased risk of perinatal death among overweight and obese women, with RRs of 1.22 and 1.55, respectively. A large part of this association (63.1%) was mediated by gestational age at delivery. These findings are interesting and I only have a few minor comments below.

1. Please add number of perinatal deaths to the abstract. 

We have added this number to the abstract: “Perinatal mortality was 0.5% (1834 pregnancies); and was elevated in overweight (adjusted odds ratio [AOR]=1.22, 95% confidence interval [CI] 1.08-1.37) and obese women (AOR=1.55, 95% CI 1.36-1.77).”

2. What is the rationale for adjusting for hypertension? Hypertension could also be on the causal pathway between maternal BMI and perinatal death. I would suggest to exclude hypertension from the multivariable model or alternatively present an additional model without hypertension, but with the other covariates.

Thank you for the comment. The order of obesity and hypertension in the proposed causal model is uncertain and likely differs for different women. Ideally, we would utilize information about the time of onset of both obesity and chronic hypertension, however, without this information either the inclusion or exclusion of hypertension in a given analysis is likely to lead to some bias. 

As per this suggestion, we have repeated our primary analyses excluding hypertension, and found little difference in results. These are now included in Table 3. 

3. Line 243-244: Has this statement been misplaced?

No, the journal requests this within the methods section. 

4. Line 295: stillbirths? Plural s

Thank you, we have corrected this typo. 

5. Did you try testing hypertension as a potential mediator in this dataset?

While we agree that testing additional mediators such as hypertension (or GDM etc) would be of interest, our objective here was to focus strictly on gestational age at delivery so we did not include additional possible mediators. Generally speaking this means that effects of these mediators are included within the ‘direct’ effects. Future analyses focussing on multiple pathways are needed; these are beyond the scope of our paper. 

6. Line 373: …..the risk of neonatal death among infants of these women.

Thank you, we have corrected this typo.

7. Please, would you be able to add analyses of stillbirth (overall, and intrapartum and antepartum), and neonatal death as well, since you have the data in Table 2? At least in an online supplement. That could have been helpful for the overall evidence base.

Thank you for the comment. We agree that such analyses add completeness and have now added them to the online supplement (Table S5) and added these analyses to the methods and results section. 

In methods line 181-183: Finally, we repeated our primary analysis for each of stillbirth, antepartum stillbirth and neonatal death separately to determine if the relationship observed for the primary outcome differed. Due to small numbers, we were unable to conduct this analysis for intrapartum stillbirth. 

In results line 252-256: Compared to the primary analyses for perinatal death, analyses restricted to stillbirth led to slightly higher odds ratios overall, while analyses for neonatal death were attenuated. Direct effects were higher, and mediation were lower for stillbirth than for neonatal death, where the total effect was entirely indirect (Table S5). Despite this, confidence intervals for all estimates were consistent with the primary analyses (Table S5). 

Unfortunately, we did not have sufficient number of outcomes for intrapartum stillbirth to allow for this analysis, but have provided it for the other three outcomes (stillbirth, antepartum stillbirth and neonatal death) in the online supplement. 

8. Considering the large study population, have you considered doing sensitivity analyses using finer BMI categories or at least looking at severe obesity separately from grade 1 obesity? Perhaps this could be added in an online supplement.

Yes, we did consider this in our original study design, as we agree that there are often meaningful differences within different strata of obese women. Unfortunately, in our cohort the proportion of women with BMI > 35 (or >40) was small, and as the outcomes were also quite rare, analyses in these subgroups did not provide meaningful estimates and we therefore have opted to present results for all obese women combined . 

 

Reviewer #2: The authors wanted to examine the possibility that earlier delivery of obese pregnancies may be attributable to increased stillbirth and neonatal death. But I think the cause and the mechanism of stillbirth and neonatal death are different. I don't suggest discussing the two together. And pregnancy complications such as GDM, Pregnancy induced hypertension，ICP can lead to stillbirth. So they should be excluded.

Thank you for the comment. As suggested here and by reviewer 1, to complement the primary analyses we have provided additional analyses for each of stillbirth and neonatal death separately in the online supplement. 

We agree that pregnancy hypertension and GDM are likely also potential mediators of the BMI  perinatal death relationship. In our paper these mediated effects are encompassed within the remaining ‘direct effect’ of BMI. 

Removal of women with these conditions would unfortunately create a selection bias, where part of the ‘direct’ (and total) effect would be blocked, and therefore we have opted to leave them in our analyses (1). A future study allowing for multiple mediators would be the best way to more completely disentangle the complex pathway between BMI and perinatal death (2) but is beyond the scope of our current analyses and available data. 

We added the following text to the conclusion line 389-392: “To better inform the pregnancy management in obese women, further studies should continue to disentangle the causal pathways under which obesity increases the risk of perinatal death, including, for example gestational diabetes and other obesity-related pregnancy complications.”

References: 

1. Hernán MA, Hernández-Díaz S, Robins JM. A structural approach to selection bias. Epidemiology. 2004 Sep;15(5):615-25. doi: 10.1097/01.ede.0000135174.63482.43. PMID: 15308962.

2. Daniel RM, De Stavola BL, Cousens SN, Vansteelandt S. Causal mediation analysis with multiple mediators. Biometrics. 2015 Mar;71(1):1–14.

 

Editor comment: 

1. Please provide additional details regarding participant consent. In the ethics statement in the Methods and online submission information, please ensure that you have specified what type you obtained (for instance, written or verbal, and if verbal, how it was documented and witnessed). If your study included minors, state whether you obtained consent from parents or guardians. If the need for consent was waived by the ethics committee, please include this information

We have revised our ethics statement (lines 188-190) as follows: Written ethics approval was obtained from the University of British Columbia - Children’s and Women’s Hospital and Health Centre of British Columbia Research Ethics Board (H18-03154). All data were fully anonymized prior to data access or analyses. 

2.Please include a caption for figure 1.

This appears on lines 213-214 of the manuscript (after figure is referenced) and is as follows: 

Figure 1: Gestational age specific perinatal death rates per 10,000 fetuses at risk by BMI class. Data are smoothed with three week moving averages.

---

## [Editor Report · Decision Letter 1]

14 Feb 2022

The association between pre-pregnancy body mass index and perinatal death and the role of gestational age at delivery

PONE-D-21-29394R1

Dear Dr. Bone,

We’re pleased to inform you that your manuscript has been judged scientifically suitable for publication and will be formally accepted for publication once it meets all outstanding technical requirements.

Kind regards,

Angela Lupattelli, PhD

Academic Editor

PLOS ONE

---

## [Editor Report · Acceptance letter]

28 Feb 2022

PONE-D-21-29394R1 

The association between pre-pregnancy body mass index and perinatal death and the role of gestational age at delivery 

Dear Dr. Bone:

I'm pleased to inform you that your manuscript has been deemed suitable for publication in PLOS ONE. Congratulations! Your manuscript is now with our production department. 

Kind regards, 

on behalf of

Dr. Angela Lupattelli 

Academic Editor

PLOS ONE